# Deep learning versus kernel learning: an empirical study of loss landscape geometry and the time evolution of the Neural Tangent Kernel

**Stanislav Fort**[1][*]    **Gintare Karolina Dziugaite**[2][*]    **Mansheej Paul**[1]
**Sepideh Kharaghani**[2]    **Daniel M. Roy**[3,4]    **Surya Ganguli**[1]
[1]Stanford University    [2]Element AI    [3]University of Toronto    [4]Vector Institute

## Abstract

In suitably initialized wide networks, small learning rates transform deep neural networks (DNNs) into neural tangent kernel (NTK) machines, whose training dynamics is well-approximated by a linear weight expansion of the network at initialization. Standard training, however, diverges from its linearization in ways that are poorly understood. We study the relationship between the training dynamics of nonlinear deep networks, the geometry of the loss landscape, and the time evolution of a data-dependent NTK. We do so through a large-scale phenomenological analysis of training, synthesizing diverse measures characterizing loss landscape geometry and NTK dynamics. In multiple neural architectures and datasets, we find these diverse measures evolve in a highly correlated manner, revealing a universal picture of the deep learning process. In this picture, deep network training exhibits a highly chaotic rapid initial transient that within 2 to 3 epochs determines the final linearly connected basin of low loss containing the end point of training. During this chaotic transient, the NTK changes rapidly, learning useful features from the training data that enables it to outperform the standard initial NTK by a factor of 3 in less than 3 to 4 epochs. After this rapid chaotic transient, the NTK changes at constant velocity, and its performance matches that of full network training in 15% to 45% of training time. Overall, our analysis reveals a striking correlation between a diverse set of metrics over training time, governed by a rapid chaotic to stable transition in the first few epochs, that together poses challenges and opportunities for the development of more accurate theories of deep learning.

The remarkable empirical success of deep learning across a range of domains stands in stark contrast to our theoretical understanding of the mechanisms underlying this same success [1]. Indeed, we are currently far from a mature, unified mathematical theory of deep learning that is powerful enough to universally guide engineering design choices. As in many other fields of inquiry, a key prerequisite to any such theory is careful empirical measurements of the deep learning process, with the scientific aim of unearthing combinations of variables that obey correlated dynamical laws that can serve as the inspiration for future theories. Indeed, a large body of work has studied, mainly in isolation, diverse and intriguing phenomenological properties, as well as extreme simplifying theoretical limits, of deep learning. In particular, we focus on 3 intertwined aspects of deep learning that have previously been studied largely in isolation: (1) the large scale structure of deep learning loss surfaces, (2) the local geometry of such loss surfaces, and (3) and the performance of linearized training methods, like the neural tangent kernel (NTK), that has gained attention through its ability to theoretically describe an infinite width low learning rate limit of deep learning in terms of kernel machines with random data-independent kernels. The fundamental goal of this work is to obtain a more integrative view of the intertwined relations between loss landscape geometry at multiple scales of organization and the

---

[*]Equal contributions. Correspondence to: `sfort1@stanford.edu`, `karolina.dziugaite@elementai.com`

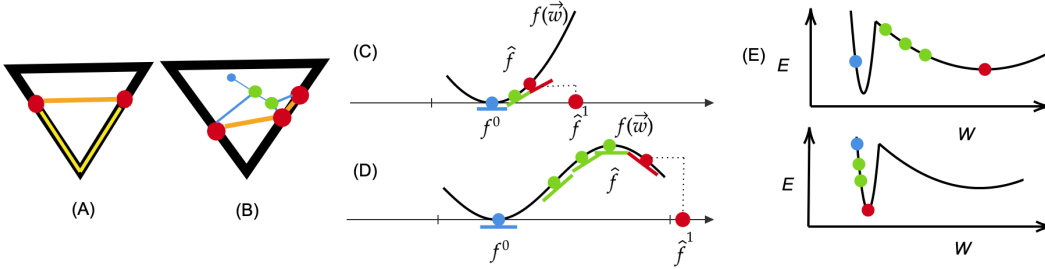

Figure 1: A conceptual overview of diverse deep learning phenomenology. (A) A schematic picture of the region of low loss (black area) in weight space as a network of high dimensional basins with lower dimensional intersections, motivated by recent work [2, 3, 4, 5, 6]. Two networks (red points) in different basins can be connected by a low loss nonlinear path (yellow) but not by a low loss linear path (orange). (B) A schematic view of the process of hierarchically exploring loss landscapes by spawning child networks [7]. A randomly initialized parent network (blue point) is trained up to a certain spawn epoch (green point) at which two (or more) child networks are spawned from with identical weights and then subsequently trained independently with different SGD minibatches (bifurcating blue lines). Two children spawned later (earlier) than a very early transition time in parent training, will arrive at the same (different) basin on the loss landscape. (C) A schematic view of NTK training. The black curve is the space of functions $f_w$ realizable by varying the parameters $w$ of a neural network and full network training proceeds along this curved function manifold (blue to green to red points). NTK training linearizes the manifold at initialization (blue point), and trains along the tangent space (blue line). Such linearized training is equivalent to kernel regression in function space where the kernel is closely related to the tangent plane along which training occurs. This panel shows a case where NTK and full nonlinear training are similar in that the kernel at initialization does not change much over learning, as shown schematically by the similar orientations of the initial (blue), intermediate (green) and final (red) tangent spaces. (D) The same as in panel (C), except now showing schematically a case where the NTK method is very different from full nonlinear training, in which the kernel changes considerably, as evidenced by the strong twisting of tangent spaces (blue, green and red lines), resulting in a final learned kernel (associated with the red tangent space) that is quite different from the initial random kernel (associated with the blue tangent space). (E) Consider an error landscape with a sharp and a wide minimum separated by an error barrier. With a small learning rate (bottom), a learning trajectory starting at an initial point (blue) will slowly descend through intermediate points (green) to a minimum position (red) in the sharp minimum, and is unable to escape it. With a larger learning rate (top), a learning trajectory that starts in the sharp minimum at a position (blue point) that is even *lower* than the error barrier, can escape the sharp minimum.

dynamics of learning in deep networks, by performing *simultaneous* measurements of many diverse properties. We describe the previous work that motivates our current measurements in Section 1, and we summarize our results and contributions in Section 8, which can be read right after Section 1.

# 1 Diverse aspects of deep learning phenomenology

**The large scale geometric structure of neural loss landscapes.** Recent work has revealed many insights into the shape of loss functions over the high dimensional space of neural network parameters. For example, [2, 3] demonstrates that training even within a random, low-dimensional affine subspace of parameter space can yield a network with low test loss. This suggests that the region of parameter space with low test loss must be a relatively high dimensional object, such that low dimensional random affine hyperplanes can generically intersect it. Moreover, [4, 8, 5] show that different, independently trained networks in weight space with low loss can be connected through nonlinear pathways (found via an optimization process) that never leave this low loss manifold. However, *direct linear* pathways connecting two such independently trained networks typically always leave the low loss manifold. The loss function restricted to such linear paths then yields a loss barrier at an intermediate point between the two networks. [6] builds and provides evidence for a unifying geometric model of the low-loss manifold consisting of a network of mutually intersecting high dimensional basins (Fig. 1A). Two networks within a basin can be connected by a straight line that never leaves the low-loss manifold, while two networks in different basins can be connected by a piecewise linear path of low loss that is forced to traverse the intersection between two basins. [9] uses these insights to argue that deep ensembles are hard to beat using local subspace sampling

methods due to the geometry of this underlying loss landscape. [7] provides further evidence for this large-scale structure by demonstrating that *after* a very early stage of training of a parent network (but not earlier) two child networks trained starting from the parameters of the parent end up in the same low loss basin at the end of training, and could be connected by a *linear* path in weight space that does not leave the low loss manifold (Fig. 1B). Furthermore, [10, 11] show that the properties of the final minimum found are strongly influenced by the very early stages of training. Taken together, these results present an intriguing glimpse into the large scale structure of the low loss manifold, and the importance of early training dynamics in determining the final position on the manifold.

**Neural tangent kernels, linearized training and the infinite width limit.** The neural tangent kernel (NTK) has garnered much attention as it provides a theoretical foothold to understand deep networks, at least in an infinite width limit with appropriate initialization scale and low learning rate [12, 13]. In such a limit, a network does not move very far in weight space over the course of training, and so one can view learning as a linear process occurring along the tangent space to the manifold of functions $f_w$ realizable by the parameters $w$, at the initial function $f^0$ (Fig. 1C). This learning process is well described by kernel regression with a certain random kernel associated with the tangent space at initialization. The NTK is also a special case of Taylorized training [14], which approximates the realizable function space $f_w$ to higher order in the vicinity of initialization. Various works compare the training of deep networks to the NTK [15, 16, 17, 18]. In many cases, state of the art networks outperform their random kernel counterparts by significant margins, suggesting that deep learning in practice may indeed explore regions of function space far from initialization, with the tangent space twisting significantly over training time, and hence the kernel being learned from the data (Fig. 1D). However, the nature and extent of this function space motion, the degree of tangent space twisting, and how and when data is infused into a *learned* tangent kernel, remains poorly understood.

**The local geometric structure of neural loss landscapes.** Much effort has gone into characterizing the local geometry of loss landscapes in terms of Hessian curvature and its impact on generalization and learning. Interestingly [19] analyses the Hessian eigenspectrum of loss landscapes at scale, demonstrating that learning leads to the emergence of a small number of large Hessian eigenvalues, and many small ones, bolstering evidence for the existence of many flat directions in low loss regions depicted schematically in Fig. 1A. [20] shows that the gradients of logits with respect to parameters cluster tightly based on the logit over training time, leading directly to the emergence of very sharp Hessian eigenvalues. Moreover, a variety of work has explored relations between the curvature of local minima found by training and their generalization properties [21, 22, 23, 24, 25, 6, 26], and how learning rate and batch size affect the curvature of the minima found [27, 28, 29], with larger learning rates generically enabling escape from sharper minima (Fig. 1E). [30] makes a connection between learning rates and the validity of NTK training, showing that for infinitely wide networks, training with a learning rate above a scale determined by the top eigenvalue of the Hessian at initialization results in a learning trajectory that outperforms NTK training, presumably by exploring nonlocal regions of function space far away from initialization.

**Towards an integrative view.** Above, we have reviewed previously distinct strands of inquiry into deep learning phenomenology that have made little to no contact with each other. Indeed, we currently have no understanding of how local and global loss geometry interacts with the degree of kernel learning in state of the art architectures and training regimes used in practice. For example, at what point in training is the fate of the final chosen basin in Fig. 1 A,B irrevocably determined? Does the kernel change significantly from initialization as in Fig. 1D? If so, when during training does the tangent kernel start to acquire knowledge of the data? Also, when does kernel learning finally stabilize? What relations do either of these times have to the time at which basin fate is determined? How does local geometry in terms of curvature change as all these events occur? Here we address these questions to obtain an integrative view of the learning process across a range of networks and datasets. While we only present results for ResNet20 trained on CIFAR10 and CIFAR100 in the main paper, in Appendix C we find similar results for a WideResNet, variations of Resnets and a Simple CNN trained on CIFAR10 and CIFAR100, indicating our results hold generally across architectures, datasets and training protocols. Many experimental details are covered in our Appendix.

## 2 Definition of measurement metrics for geometry and training

We now mathematically formalize the quantities introduced in the previous section as well as define more quantities whose dynamics we will measure during training. Let $S = ((x_i, y_i), 1 \leq i \leq m)$ be $m$ training examples, with $y_i \in \{0, 1\}^K$, where $K$ is the number of classes. Let $f_w(x)$ denote the $K$-dimensional output vector of logits, of a neural network parameterized by weights $w \in \mathbb{R}^d$ on

input $x$. We are interested in the average classification error $R_S^{0-1}(w)$ over the samples $S$. For training purposes, we also consider a (surrogate) loss $\ell(\hat{y}, y)$ for predicting $\hat{y}$ when the true label is $y$. Denote by $g(\hat{y}, y)$ the gradient of $y' \mapsto \ell(y', y)$, evaluated at $y' = \hat{y}$. Write $g_w(S)$ for concatenation of the gradient vectors $g(f_w(x_i), y_i)$, for $i = 1, \ldots, m$. Let $J_w(x) \in \mathbb{R}^{K \times d}$ be the Jacobian of $f_w(x)$ with respect to the parameters $w$. Define $J_w(S) \in \mathbb{R}^{mK \times d}$ to be the concatenation of $J_w(x_1), \ldots, J_w(x_m)$, which is then the Jacobian of $f_w(S)$ with respect to the parameters $w$. The $k^{th}$ row of $J_w(x)$, denoted $(J_w(x))_k$, is a vector in $\mathbb{R}^d$. Let $H_w(x)$ be the $K \times d \times d$ tensor where $(H_w(x))_k = \nabla_w (J_w(x))_k \in \mathbb{R}^{d \times d}$ is the Hessian of logit $k$ w.r.t. weights $w$.

**Training Dynamics, Linearized training, and introduction of a data-dependent NTK.** Let $(w_t)_{t \in \mathbb{N}}$ be the weights at each iteration of SGD, based on minibatch estimates of the training loss $\hat{R}_{\bar{S}}(w) = \frac{1}{n} \sum_{x_i \in \bar{S}} \ell(f_w(x_i), y_i)$, where $\bar{S} \subset S$ is a subsample of data of size $n$. We write $f_t(x)$ for $f_{w_t}(x)$ and similarly for $g_t$, $J_t$, and $H_t$. The SGD update with learning rate $\eta$ is then

$$\Delta_t := w_{t+1} - w_t = -\eta \nabla_w R_{\bar{S}}(w_t), \tag{1}$$

Consider also a second-order Taylor expansion to approximate the change to the logits for input $x_i$:

$$f_{t+1}(x_i) \approx f_t(x_i) - J_t(x_i)\Delta_t + \|\Delta_t\|_{H_t(x_i)}, \tag{2}$$

where

$$\|\Delta_t\|_{(H_t(x_i))_k^T} := \langle \Delta_t, (H_t(x_i))_k^T \Delta_t \rangle. \tag{3}$$

Note, that for an infinitesimal $\eta$, the dynamics in Eq. (1) are those of gradient flow, and terms higher than order 1 in Eq. (2) vanish. In this case, steepest descent in the parameter space corresponds to steepest descent in the function space using a *neural tangent kernel* (NTK),

$$\kappa_t(x, x') = J_t(x)J_t(x')^T. \tag{4}$$

Let $\kappa_t(S)$ denote the $m$ by $m$ gram matrix with $i, j$ entry $\kappa_t(x_i, x_j)$. If $\kappa_t(S) = \kappa_{t_0}(S)$ for $t > t_0$, i.e., if the tangent kernel is constant over time, then the dynamics correspond to those of training the neural network linearized at time $t_0$. The kernel has been shown to be nearly constant in the case of very wide neural networks at initialization (see, e.g., [12, 31, 32, 33, 16, 34]). Intuitively, we can think of each of the $d$ columns of $J_w(x) \in \mathbb{R}^{K \times d}$ as a tangent vector to the manifold of realizable neural network functions in the ambient space of all functions of $K$ logits over input space $x$, at the point $f_w(x)$ in function space. Thus the span of the $d$ columns of $J_w(x)$, as $x$ varies, constitute the tangent planes in function space depicted schematically in Fig. 1CD. Since the kernel is the Gram matrix associated with these tangent functions, evaluated at the training points, then if the tangent space twists substantially, the kernel necessarily changes (as in Fig. 1D).

Conversely, if the NTK does not change substantially from initialization, then the full SGD training can be well approximated by training along the tangent space to $f_0$ at initialization, yielding the linearized training dynamics. This approach can be generalized to training along higher order Taylor approximations of the manifold $f_w(x)$ in the vicinity of the initial function $f_0$ [14]. In this work, in order to explore function space geometry and its impact on training, we extend this approach by doing full network training up to time $\tilde{t}$, and then linearized training subsequently. This yields a linearized training trajectory $\{w_t^{\tilde{t}}\}_{t=\tilde{t}}^T$, which can then be compared to the weight dynamics under full training (see Appendix for details). This approach geometrically corresponds to training along an intermediate tangent plane (one of the green planes in Fig. 1CD), or equivalently, corresponds to learning with a *data-dependent* NTK. This novel examination of how much training time is required to learn a high performing NTK, distinct from the random one used at initialization, and relations between this time and both the local and large scale structure of the loss landscape, constitutes a key contribution of our work.

**Hierarchical exploration of the loss landscape through parents and children.** In order to explore the loss landscape and the stability of training dynamics in a more multiscale hierarchical manner than is possible using completely independent training runs, we employ a method of parent-child spawning [7] (shown schematically in Fig. 1B). In this process, a parent network is trained from initialization to a spawning time $t_s$, yielding a parent weight trajectory $\{w_t\}_{t=0}^{t_s}$. At the spawn time $t_s$, several copies of the parent network are made, and these so-called children are then trained with independent minibatch stochasticity, yielding different child weight trajectories $\{w_t^{t_s, a}\}_{t=t_s}^T$, where $a$ indexes the children, and $T$ is the final training time. We will be interested in various measures of the distance between children after training, as a function of their spawn time $t_s$, as well as measures of the distance between the same network (either parent or child) at two different training times. We turn to these various distance measures next.

**Kernel distance.** For finite width networks, the kernel $\kappa_t(S) = \kappa_{w_t}(S)$ changes with training time $t$. We compare two Kernel gram matrices in a scale-invariant manner by computing a *kernel distance*:

$$S(w, w') = 1 - \frac{\text{Tr}(\kappa_w(S)\kappa_{w'}^T(S))}{\sqrt{\text{Tr}(\kappa_w(S)\kappa_w^T(S))}\sqrt{\text{Tr}(\kappa_{w'}(S)\kappa_{w'}^T(S))}}.$$

**Kernel velocity.** We further track the speed at which the kernel changes. As discussed above, in non-linear neural networks, we do not expect Eq. (3) to vanish. In order to capture the evolution of the quantity in Eq. (3), we compute the *kernel velocity* $v(t) \equiv S(w_t, w_{t+dt})/dt$, i.e. the rate of change of kernel distance. We use a time separation of 0.4 epochs to capture appreciable change.

**Error barrier between children.** To assess (and indeed *define*) whether two children arrive at the same basin or not at the end of training (see e.g. Fig. 1AB), we compute the error barrier between children along a *linear* path interpolating between them in weight space. Let $w_t^\alpha = \alpha w_t + (1-\alpha)w_t'$, where $w_t'$ and $w_t$ are the weights of two child networks, spawned from some iteration $t_s$, and $\alpha \in [0,1]$. At various $t_s$ we compute $\max_{\alpha \in [0,1]} \hat{R}_S(w_t^\alpha) - \frac{1}{2}(\hat{R}_S(w_t) + \hat{R}_S(w_t'))$, which we call the *error barrier*. Note, that the error barrier at the end of training between two children is the same as *instability* in [7].

**ReLU activation pattern distance.** In a ReLU network, the post-nonlinearity activations in layer $l$ are either greater or equal to 0. We can thus construct a tensor $B_w(S)$, with $(B_w(S))^{i,j,l} = 1$ if for an input $x_i$, $j^{\text{th}}$ node in the $l^{\text{th}}$ layer is strictly positive, and $(B_w(S))^{i,j,l} = 0$ otherwise. We compare ReLU on/off similarity between networks parameterized by $w$ and $w'$ by computing the Hamming distance between $B_w(S)$ and $B_{w'}(S)$, and normalizing by the total number of entries in $B_w(S)$.

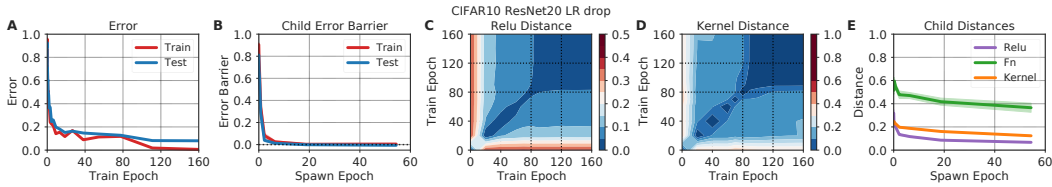

Figure 2: SOTA ResNet20 trained on CIFAR10 using SGD with momentum and learning rate drops.

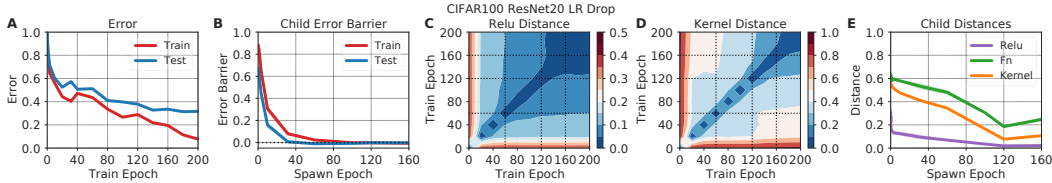

Figure 3: ResNet20 trained on CIFAR100 using SGD with momentum and learning rate drops.

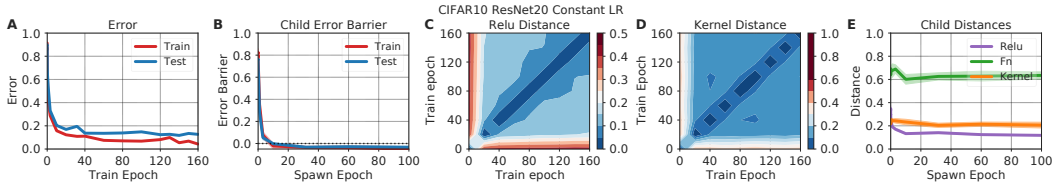

Figure 4: ResNet20 trained on CIFAR10 using SGD with momentum and constant learning rate.

Figures 2 to 4: An integrated view of learning. (A) Parent network learning curves. (B) Error barrier between pairs of children at the end of training, as a function of spawn time, with children trained for same number of epochs as the parent. (C) and (D) Heatmaps representing the ReLU and kernel distance between a parent network at different pairs of training times. Dashed black lines indicate epochs at which the learning rate is dropped. (E) ReLU, function space, and kernel distances between pairs children at the end of training, as a function of spawn time.

**Function space distance.** To compute the distance between the two functions $f_w$ and $f_{w'}$, parameterized by weights $w$ and $w'$, we would ideally like to calculate the degree of disagreement between their outputs averaged over the whole input space $x$. However, since this is computationally intractable, we approximate this distance by the normalized fraction of test examples on which their predicted labels

disagree. Let $S^{\text{test}}$ denote the test set. Then, $\|f_w(x) - f_{w'}(x)\|_{S^{\text{test}}} = \frac{1}{Z|S^{\text{test}}_x|} \sum_{x \in S^{\text{test}}_x} (f_w(x) \neq f_{w'}(x))\,1$ where $S^{\text{test}}_x$ are test inputs and $Z$ is a normalizing constant chosen to aid comparison. In particular, we define $Z$ to be the expected number of examples on which two classifiers would disagree assuming each made random independent predictions with the same error rates, $p$ and $p'$, as their error rates on the test set. This quantity is used also by [9], and is given by $Z = p(1-p') + p'(1-p) + pp'\frac{K-2}{K-1}$, where $K$ is the number of classes. A unit distance indicates two networks make uncorrelated errors.

## 3  An integrative view of learning dynamics

Figs. 2 and 3 plot the full range of metrics defined in Section 2 for two SOTA networks. Panel A presents standard training curves. Panel B confirms the results of [7], that the error barrier on a linear path between two children decreases rapidly with spawning time, falling close to 0 within two to five epochs. Panel C and D indicate that the NTK changes rapidly early in training, and more slowly later in training, as quantified by ReLU activation distance (C) and kernel distance (D) measured on a parent run at different pairs of times in training. Finally, Panel E shows that function, kernel and ReLU distances between children at the end of training also drop as a function of spawn time.

We note that the SOTA training protocols in Figs. 2 and 3 involve learning rate drops later in training, which alone could account for a slowing of the NTK evolution. Therefore we ran a constant learning rate experiment in Fig. 4. We see that all tracked metrics still exhibit the same key patterns: the error barrier drops rapidly within a few epochs (B), the NTK evolves very rapidly early on, but continues to evolve at a constant slow velocity later in training (C,D), and final distances between children drop at an early spawn time and remain constant thereafter (E).

Overall, these results provide an integrative picture of the learning process, which reveals an early, extremely short chaotic period in which the final basin chosen by a child is highly sensitive to SGD noise and the NTK evolves very rapidly, followed by a later more stable phase in which the basin selection is determined, the NTK continues to evolve, albeit more slowly, and the final distance between children remains smaller. In the next few sections we explore these results in more detail.

Equivalent results for other networks are shown in the Appendix in Figures Figs. 12 to 17, together with additional network properties tracked over epochs.

## 4  The local and global geometry of the loss landscape surrounding children

We first explore how both the global and local landscape geometry surrounding two child pairs and their spawning parent depend on the spawn time $t_s$ in Fig. 5. These three networks define a 2D affine plane in weight space and a curved 2D manifold in function space. The first two columns of Fig. 5 clearly indicate that two children spawned at an early time $t_s$ in the chaotic training regime arrive at two different loss basins that are well separated in function space (top row), while two children spawned at a later time $t_s$ in the stable training regime arrive at the same loss basin, though this loss basin can still exhibit non-negligible function diversity (albeit smaller than the diversity between basins). Furthermore, the right two columns of Fig. 5 indicate that the test error as a function of position along the tangent plane to the 2D curved manifold in function space (either at the spawn point or a child point) is insufficient to describe the error along the full curved 2D manifold in function space when the children are in different basins (top row), but can approximately describe the loss landscape when the children are in the same basin (bottom row). Thus Fig. 5 constitutes a new direct *data-driven* visualization of loss landscape geometry that provides strong evidence for several aspects of the conceptual picture laid out in Fig. 1: the existence of multiple basins (Fig. 1A), the chaotic sensitivity of basin fate to SGD choices early in training (Fig. 1B), and the twisting of tangent planes in function space that occur as one travels from one basin to another (Fig. 1ACD). See also Fig. 11 for a t-SNE visualization of the bifurcating evolution of all parents and children in function space that further corroborates this picture of loss landscape geometry and its impact on training dynamics.

In Fig. 6 we explore more quantitatively the relationship between the final function space distance between children, spawn time of children, and the error barrier. This figure demonstrates that the error barrier drops to zero rapidly within 2-3 epochs, and then after that, the later two children are spawned, the closer they remain to each other. Since these experiments were done by training parents and children at a constant learning rate over 200 epochs such that child distances stabilized, the reduction in achievable function space distance between children as a function of spawn time cannot be explained either by learning rate drops or by insufficient training time for children (see Fig. 10).

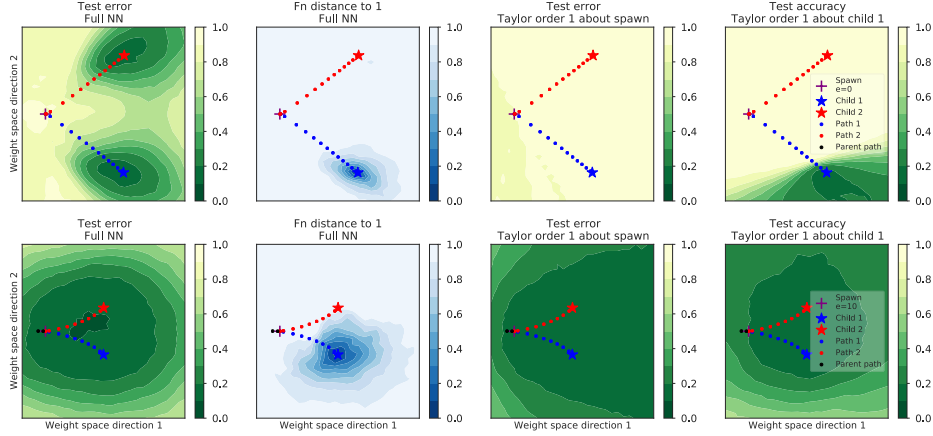

Figure 5: The error landscape and function space geometry on a 2D section defined by a pair of children (red and blue stars) and the spawning parent (purple cross) when the spawn point is in the early chaotic (top row) and late stable (bottom row) regimes of training. All other training points are projected to this 2D section. The left two columns show, as a function of position on this 2D section, the test error and the function space distance to a chosen child (blue star). The right two columns show the test error along an affine tangent plane in function space obtained by a first order Taylor expansion of $f_w$ in weight space around the weights of two different networks (the spawning parent and one of the children). A function space point along the tangent plane at $f_w$ is identified with a point on the curved 2D section in function space through the relation $f_w + \Delta w \cdot \nabla_w f_w \to f_{w+\Delta w}$.

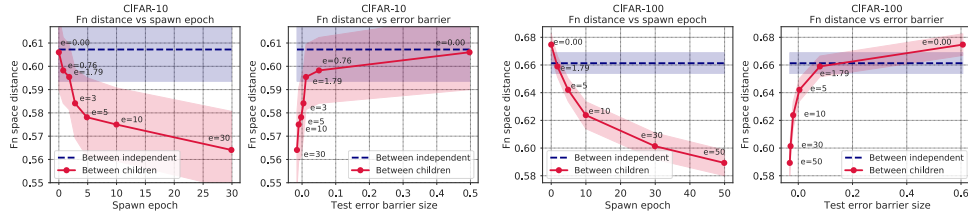

Figure 6: Relation between error barrier and child function distance for ResNet20 on CIFAR 10 and 100. Left panels show how final child distance (near 200 epochs) falls off with *spawn* epoch (red curve). The purple baseline indicates final distance between two independent parents. Right panels plot function distance as a function of error barrier. See also Fig. 10 for detailed evolution of both quantities with training rather than spawn epoch. Error bars reflect std. dev. across the last 25 epochs. The function prediction embeddings are shown in Fig. 11.

## 5 NTK velocity slows down and stabilizes after basin fate is determined

We next explore the relation between error barrier and kernel velocity in Fig. 7 by zooming in on the early epochs, compared to the full training shown in Fig. 2-3 panels B and D. This higher resolution view clearly reveals that the early chaotic training regime is characterized by a tightly correlated reduction in both error barrier and kernel velocity, with the latter stabilizing to a low non-zero velocity after the error barrier disappears. Thus the NTK evolves relatively rapidly until basin fate is sealed.

## 6 The data-dependent NTK rapidly learns features useful for performance

The rapid evolution of the NTK during the chaotic training regime and its subsequent constant velocity motion after basin fate determination, as shown in Fig. 7, raises a fundamental question: at what point during training does the NTK learn useful features that can yield high task performance, or even match full network training? We answer these questions in Fig. 8 through a two step training protocol. We first train the full nonlinear network up to a time $\tilde{t}$. We then Taylor expand the full nonlinear network $f_w$ obtained at time $\tilde{t}$ with respect to the weights $w$, and perform linearized training thereafter up to a total time $T$. Geometrically this corresponds to training for time $\tilde{t}$ up to one of the intermediate green points in Fig. 1CD, and then subsequently training only within the green tangent

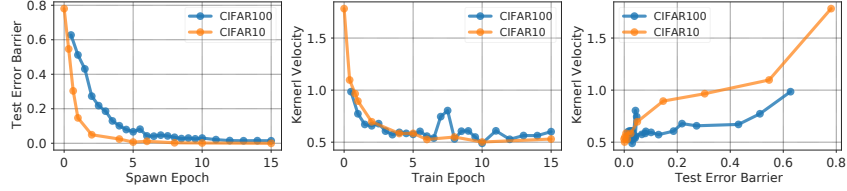

Figure 7: Relation between test error barrier and kernel velocity for a ResNet20 trained on CIFAR10 and CIFAR100. Both the test error barrier between children (left) and the kernel velocity of the parent (middle) fall off and stabilize early in time and exhibit strongly correlated dynamics (right).

space about that point in function space. We can think of this as training with a data-dependent NTK kernel that has been learned using data from time 0 to $\tilde{t}$. Classic NTK training corresponds to a random kernel arising when the onset time $\tilde{t}$ of linearized training is 0.

Using this two step procedure, Fig. 8 demonstrates several key findings. First, extremely rapidly, within $\tilde{t} = 3$ to 4 epochs, the data dependent NTK has learned features that allow it to achieve significantly better performance (i.e. error drops by at least a factor of 3) compared to the classic NTK obtained at initialization (see rapid initial drop of green curves in Fig. 8). Second, by about $\tilde{t} = 30$ to 90 epochs, representing 15% to 45% of training, the data-dependent NTK essentially matches the performance of a network trained for the standard full 200 epochs (compare green curves to purple baseline in Fig. 8). This indicates that the early chaotic training period characterized by rapid drops in error barrier and kernel velocity in Fig. 7 also corresponds to rapid kernel learning: useful information is acquired *within a few epochs*. This kernel learning continues, albeit more slowly after the initial chaotic learning period is over and the basin fate is already determined.

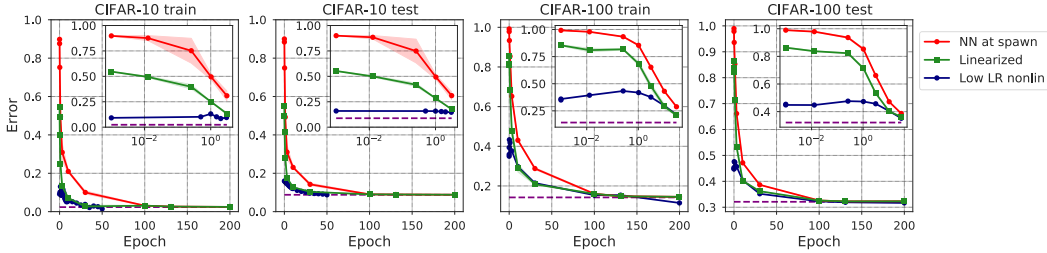

Figure 8: Linearized training vs. ordinary training. The red baseline curves show the (test/train) error of a network using full nonlinear training as a function of training epoch $\tilde{t}$. The dashed purple constant baseline, purely for reference, indicates the error obtained at epoch $T = 200$. The green line indicates the error of a data dependent NTK obtained at training epoch $\tilde{t}$; i.e. the error for the green line is obtained by full nonlinear network training up to time $\tilde{t}$, and then subsequent linearized training with an optimal early stopping criterion for the test error. The train/test error at the optimal stopping time is plotted as a function of the *onset* time $\tilde{t}$ of linearized training, reflecting the performance of the data-dependent NTK at time $\tilde{t}$. The blue curve is obtained identically to the green curve, except instead of using linearized training, we use full nonlinear training at the lowest possible learning rate after time $\tilde{t}$, that still ensures convergence after 1000 epochs. We explore the gap between the green and blue curves in Fig. 9. In the Appendix in Fig. 19 we show additional results for WideResNet and in Fig. 18 Taylor expansions of order 2 for ResNet.

## 7 NTK and nonlinear training remain different even at low learning rates

In the NTK limit, which involves *both* infinite widths and infinitesimal learning rates, linearized training and full nonlinear training dynamics provably coincide. However, the persistent performance gap up to 30 to 90 epochs between linearized and full nonlinear training (green curves versus purple baselines in Fig. 8) indicates the NTK limit does not accurately describe training dynamics used in practice, at finite widths and large learning rates. We remove one of the two reasons for the discrepancy by comparing the same linearized training dynamics to extremely low learning rate nonlinear training dynamics (blue curves in Fig. 8). In this finite width low learning rate regime, we find, remarkably, that a significant performance gap persists between linearized and nonlinear training

(the red nonlinear training advantage region in Fig. 9, left), but only during the first few epochs of training, corresponding precisely to the chaotic regime before basin fate is sealed. Indeed the disappearance of this low learning rate nonlinear advantage is tightly correlated with the disappearance of the error barrier (Fig. 9, right). This indicates that while the data-dependent NTK limit can describe well the low (but not high) learning rate dynamics after the first few epochs, this same NTK limit cannot accurately describe the full nonlinear learning dynamics during the highly chaotic early phase prior to basin fate determination, *even when* the full nonlinear training uses very low learning rates, *and when* the NTK is learned from the data. We present additional experiments with Taylor expansions of order 2 on ResNet in Fig. 18 and linear order for WideResNet in Fig. 19.

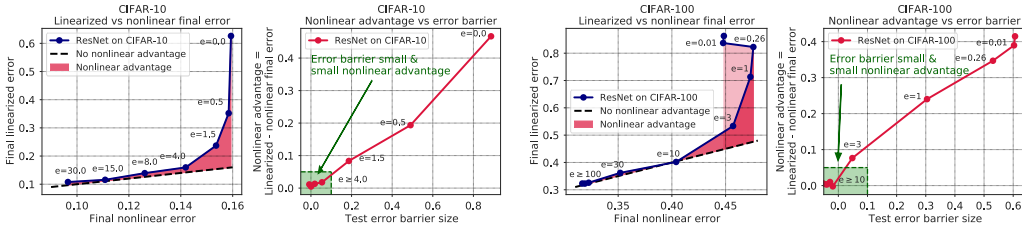

Figure 9: Relation between low learning rate nonlinear error advantage and error barrier size. For each dataset, the left panel blue curves plot the error obtained by linearized training (green curve in Fig. 8) against the error obtained by nonlinear low learning rate training (blue curves in Fig. 8), with the epoch indicating the onset time $\tilde{t}$ of both. The dashed line is the unity line, and so the height of the red region indicates the nonlinear advantage, or error reduction obtained by low learning rate nonlinear training relative to linearized training. Right panels plot this nonlinear advantage against error barrier size.

## 8   Summary of contributions and discussion

In summary we have performed large scale *simultaneous* measurements of diverse metrics (Figs. 2 to 4) finding a strikingly universal chaotic to stable training transition across datasets and architectures that completes within two to three epochs. During the early chaotic transient: (1) the final basin fate of a network is determined (Fig. 7 left); (2) the NTK rapidly changes at high speed (Fig. 7 middle and Fig. 4D); (3) the NTK rapidly learns useful features in training data, outperforming the standard NTK at initialization by a factor of 3 within 3 to 4 epochs (Fig. 8 green curves); (4) even low learning rate training retains a nonlinear performance advantage over linearized NTK training with a learned kernel (Fig. 9 red regions); and (5) the error barrier, kernel velocity, and low learning rate nonlinear advantage all fall together in a tightly correlated manner (Fig. 7, right) and (Fig. 9, right). After this rapid chaotic transient, training enters a more stable regime in which: (6) SGD stochasticity allows more limited child exploration in terms of function space distance, leading to smaller function diversity within basins compared to between basins (Figs. 5 and 6); (7) the kernel velocity stabilizes to a fixed nonzero speed (Fig. 7 middle and Fig. 4D); (8) the data dependent kernel performance continues to improve, matching that of full network training by 30 to 90 epochs, of training, representing 15% to 45% of the full 200 epochs (Fig. 8 green curves).

The empirical picture uncovered by our work is much richer than what any theory of deep learning can currently capture. In particular, the NTK theory attempts to describe the entire nonlinear deep learning process using a *fixed random* kernel at initialization. While this description is provably accurate at infinite width and low learning rate, our results show it is a poor description of what occurs in practice at finite widths and large learning rates (Figs. 7 and 8). More interestingly, the NTK theory is even a poor description of nonlinear training at finite width and extremely low learning rates, especially during the early chaotic training phase (Fig. 9).

This rich phenomenological picture of the rapid sequential nature of the learning process could potentially yield practical dividends in terms of a theory for the rational design of learning rate schedules. For example, the timing of optimized learning rate drops coincide with the time when the data-dependent tangent kernel can achieve high accuracy. Indeed our observations are consistent with findings in [11]. But more generally, we hope that our empirical measurements of such a rich phenomenology may serve as an inspiration for developing an equally rich unifying theory of deep learning that can simultaneously capture these diverse phenomena.

## Broader Impact

The goal of our work is to gain a better understanding of deep neural networks. This could potentially make machine learning applications more reliable and transparent in the long run.

## Funding Sources

DMR was supported, in part, by an NSERC Discovery Grant, Ontario Early Researcher Award, and a stipend provided by the Charles Simonyi Endowment. SG thanks the Simons Foundation, James S. McDonnell Foundation, NTT Research, and an NSF Career award for support. This research was in part carried out while GKD and DMR participated in the Special Year on Optimization, Statistics, and Theoretical Machine Learning at the Institute of Advanced Studies.

## Acknowledgements

The authors would like to thank Jonathan Frankle and Mufan Li for feedback on drafts, and Shems Saleh for helping to produce Fig. 1.

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
