[Supplementary Material]

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

# A    Function distance between children runs

The function distance between the children runs is shown in Fig. 10 (right column) for ResNet20 on CIFAR-10 (top row) and CIFAR-100 (bottom row). The signal is relatively noisy from iteration to iteration, we therefore overlay the raw data with a smoothed out version with a window of $\pm 5$ epochs in Fig. 10.

We also produced a t-SNE [35] visualization of parent and children evolution in the function space. To do that, we took predictions of the parent and children runs at different stages of their training on the test set. We then flattened the vector of predicted probabilities for all images and all of their classes into a single long vector, one for each stage of training of a network. We then used the t-SNE embedding to embed all parent and children runs into a 2D space. For the individual panels in Fig. 11, we highlighted the relevant embedded points, however, due to the nature of t-SNE all predictions had in fact been embedded together.

# B    Definitions of additional metrics

## B.1    Logit gradient centroid alignment

The logit gradient centroid for each class $k$ is defined as $\mu_w^k = \frac{1}{m} \sum_{i=1}^m \nabla_w (f_w(x_i))_k$. Previous work [20, 19] has shown that the span of the $K$ logit gradient centroids approximately tracks an important local quantity: the span of the top $K$ directions of maximal Hessian curvature (see also see Appendix B.2). Thus the span of the logit gradient centroids describes the orientation of the walls of the basins shown schematically in Fig. 1AB, and two trained networks with highly dissimilar logit gradient centroids likely lie in differently oriented basins. In order

Figure 10: The test accuracy and function space distance between independent runs and children spawn at different epochs for ResNet20 on CIFAR-10 (top row) and CIFAR-100 (bottom row). The noisy curves show the raw data recorded every $1/3$ of an epoch, while the thick lines show the moving average of $\pm 5$ epochs.

Figure 11: A visualization of the function space motion during training of 3 parents and 3 pairs of children spawned at early, late and intermediate times in a common t-SNE embedding. Different networks are highlighted in different plots from left to right. Siblings spawned later remain closer to each other and to their parent at the end of training.

to evaluate how logit gradient centroids compare at $w$ and $w'$, we compute average cosine similarity

$$\frac{1}{K} \sum_{k=1}^{K} \mu_w^k \cdot \mu_{w'}^k / (\|\mu_w^k\| \|\mu_{w'}^k\|), \tag{5}$$

which we refer to as logit gradient centroid alignment.

## B.2 Logit gradient centroids and the top Hessian eigenvectors

Here we note qualitative relations between the space of logit gradient centroids and the principal Hessian subspace. First, without loss of generality, assume $K = 1$ and consider a single data point $(x, y)$. Consider mean squared error (MSE) loss, with empirical risk term $R_S(w) = (f_w(x) - y)^2$. Then

$$\frac{\mathrm{d}^2 R_S(w)}{\mathrm{d}w_i \mathrm{d}w_j} = \frac{\mathrm{d}f}{\mathrm{d}w_i} \frac{\mathrm{d}f}{\mathrm{d}w_j} + \left[ (f_w(x) - y)(H_w)_{i,j} \right], \tag{6}$$

where $H_w = \nabla^2 f_w(x)$ (as defined in Section 2).

If $f_w = f_{w_0} + \langle \nabla f_{w_0}, w - w_0 \rangle$, then $H_w$ is zero and the second term vanishes. Similarly, for $K \in \mathbb{N}$ and a training set $S$ of size $m$, the Hessian of MSE error loss is equal to $\nabla^2 R_S(w) = \frac{1}{K^2 m} \sum_{k,k',x \in S_x} (J_{w_0}(x))_k (J_{w_0}(x))_{k'}^T$.

Since $(J_w(x))_k = \nabla_w (f_w(x))_k$, and logit gradient centroids are defined as

$$\mu_w^k = \frac{1}{m} \sum_{i=1}^{m} \nabla_w (f_w(x_i))_k, \tag{7}$$

we decompose $(J_w(x))_k = \mu_w^k + e_w^{n,k}$, for $w = w_0$. This decomposition of $J_w$ has been previously studied in [20, 19]. Then $\nabla^2 R_S(w) = \frac{1}{K^2 m} \sum_{k,k'} \mu_w^k (\mu_w^{k'})^T + \mathcal{O}(e)$.

According to our empirical results as well as other literature [36, 20], the logit gradient centroids are mutually almost orthogonal. Therefore, $\nabla^2 R_S(w) \approx \sum_k \|\mu_w^k\|^2$. For mutually orthogonal gradient centroids $\mu_w^k$, this amounts to a singular value decomposition with $K$ non-zero singular values $\|\mu_w^k\|^2$ associated with singular vectors $\mu_w^k$.

While the relative length of $\mu_w^k$ can be changing with training time, the empirically observed stability of their directions $\mu_w^k$ makes the Hessian eigenvector associated with the highest eigenvalue constrained to lie primarily within the vector space defined by the span of $\{\mu_w^k\}_{k \in \{1,\dots,K\}}$. This subspace of dimension $K$ ($K = 10$ for CIFAR-10, $K = 100$ for CIFAR-100) has a significantly lower dimensions than the typical weight space of the network.

When evaluating the cosine similarity of the logit gradient centroids as in Eq. (5), we are approximately estimating the overlap between the low dimensional subspaces to which the sharpest directions of the Hessian are constrained between the two networks.

## B.3 Escape threshold

Consider iterative optimizers that at each iteration perform an update on the weights of the form

$$w_{t+1} = w_t - \eta \Delta_t(\bar{S}), \tag{8}$$

where $\bar{S}$ is a minibatch, i.e., a random subset $\bar{S} \subset S$.

Let $L_t = \nabla^2 R_S(w_t)$ be the Hessian of the empirical loss. Then, by a second order Taylor expansion of $R_S(w_{t+1})$ around $w_t$, we have

$$R_S(w_{t+1}) - R_S(w_t) \approx \frac{1}{2}\eta^2 \langle \Delta_t(\bar{S}), L_t \Delta_t(\bar{S}) \rangle - \eta \langle \Delta_t(\bar{S}), \nabla R_S(w_t) \rangle. \tag{9}$$

The loss after one iteration decreases if the difference $R_S(w_{t+1}) - R_S(w_t)$ is negative. Under the second order approximation, the condition for non-increasing loss is equivalent to

$$\eta \frac{1}{\|\nabla R_S(w_t)\|^2} \left( \langle \Delta_t(\bar{S}), L_t \Delta_t(\bar{S}) \rangle - 2 \langle \Delta_t(\bar{S}), \nabla R_S(w_t) \rangle \right) \leq 0. \tag{10}$$

We refer to left hand side term in Eq. (10) as the *escape threshold*. If the escape threshold is below zero, then the trajectory will be descending in the quadratic basin, under the assumption that the local quadratic approximation is accurate. If the escape threshold is positive, the loss will increase or the trajectory will escape the quadratic basin.

Gradient descent (GD) update is $\Delta_t(S) = \nabla R_S(W_t)$. Combining this with the bound $\langle \Delta_t, L_t \Delta_t \rangle \leq \lambda_t \|\Delta_t\|^2$, where $\lambda_t$ is the spectral norm of $L_t$, Eq. (10) simplifies to $2 - \eta \lambda_t$. We refer to this term as the escape threshold for GD.

# C  Additional results

Here we present additional experiments similar to the ones in Figs. 2 to 4 that include measurements of logit gradient centroid clustering, Hessian spectral norms and escape-time threshold analysis (for the networks with no batch norm).

## C.1  Diverse metrics for loss landscape and training are highly correlated

We compute all of above metrics for SOTA networks in Figs. 2, 3 and 14 to 16 (see Appendix D for details of training and hyperparameters). The top rows of Figs. 14 to 16 describe the dynamics of parent training. From left to right, the test and training error drop, as well as the top Hessian eigenvalue drops over time, and all three distances (ReLU pattern, mean logit gradient, and kernel distance) computed on a parent run between pairs of training epochs reveal a rapid change around a very early point at about 5 epochs, followed by slow freezing of all three quantities. In particular, there is significant kernel learning. Moreover, there is a period of time early on where the top Hessian eigenvalue $\lambda > 2/\eta$ where $\eta$ is the learning rate. This large learning rate condition would be necessary for gradient descent to escape a quadratic minimum (Fig. 1E, top).

The bottom row computes various distances between pairs of children at the *end* of training as a function of the common time $t_s$ at which the pairs of children were spawned. These plots indicate that at an *extremely* early spawn epoch (x-axis) of around $t_s = 1$, the basin fate of the two children is sealed by their parent. Beyond $t_s = 1$, the two children end up in the same basin, as evidenced by the lack of a loss barrier along a linear path between them, and are much closer to each other, as measured by distances in (from left to right), weight space, ReLU pattern, logit gradient, function space, and kernel space. In contrast, before this early spawn epoch of 1, the final basin choice of the children displays a chaotic sensitivity to SGD steps, as evidenced by a large loss barrier and larger distances. Intriguingly, significant, albeit slow kernel learning continues after basin fate selection.

## C.2  Further discussion

The integrative analysis of diverse measurements during neural network training (Figs. 2, 3 and 14 to 17) and the results of linearized training (Fig. 8) reveal a uniform and striking story. First, very early in the training process—between 1 and 10%—of training time, the final basin fate of the neural network is determined. After this point, large scale motion of the neural network is no longer influenced by SGD noise and the networks trained with different SGD noise converge to low loss points in the same (linearly connected) basin. This is indicated by the fact that children spawned after this point are linearly connected through low error networks: the error barrier between spawned children goes below 0 at around 10 epochs for all networks. Additionally, the escape threshold and spectral norm analysis reveals that, after this point, the learning rate is small enough to keep the network within the local quadratic approximation of the loss surface, further supporting the hypothesis that the network does not leave the basin. Once the network is in the basin, we find that various metrics of network distance start decreasing.

Our linearized training analysis (Figs. 18 and 19) reveals further interesting features about this early period of training. Very early in the training process, before an epoch is completed, the data-dependent NTK—the first order approximation of the neural network—rapidly starts learning useful information about the data as evidenced by the fall off of the green lines in Figs. 8, 18 and 19. This occurs uniformly across networks and datasets. When a kernel machine is trained with these data dependent features, it performs significantly better

Figure 12: ResNet20 with batchnorm trained on CIFAR10 using momentum.

Figure 13: ResNet20 with batchnorm trained on CIFAR100 using momentum.

Figure 14: SimpleCNN trained on CIFAR10 using Adam.
Grey dashed lines in F–J represent a straight line between the y axis values at epoch 0 and the final epoch. X-axis in F–J indicates the spawning epoch. Dashed lines in C–E mark epochs when the learning rate was dropped. In A, 'child' line represents the final test error of a child spawn at epoch indicated on the x-axis. Fig. 12B, Fig. 13B, and Fig. 14B depicts the spectral norm of the Hessian compared to the learning rate $\eta$. *These are additional results extending Figs. 2 to 4*

than the NTK at random initialization. In fact, less than halfway through training, the data-dependent neural tangent kernel machine performs nearly as well as as the full network.

These observations have two important implications. First, while training the NTK at random initialization may not be very representative of training finite sized networks, across a range of networks, the data-dependent NTK obtained from a small amount of training is. Second, the features built up by the NTK relatively early in the training process are sufficient for achieving low errors competitive with the full non-linear networks.

While deep learning research often focuses on improving accuracy towards the end of training, our results show that the early phase of training is important for determining the final fate of the network. A better understanding of this phase may provide us with tools to diagnose and improve networks early on in the training process, thus decreasing the cost of training neural networks.

Figure 15: WideResNet-16-4 trained on CIFAR10 using momentum.

Figure 16: ResNet20 without batchnorm trained on CIFAR10 using Adam.

Figure 17: ResNet20 without batchnorm trained on CIFAR100 using Adam.

Grey dashed lines in F–J represent a straight line between the y axis values at epoch 0 and the final epoch. X-axis in F–J indicates the spawning epoch. Dashed lines in C–E mark epochs when the learning rate was dropped. In A, 'child' line represents the final test error of a child spawned at epoch indicated on the x-axis. Fig. 15B depicts the spectral norm of the Hessian compared to the learning rate $\eta$. Fig. 16B and Fig. 17B look at the escape threshold (See Appendix B.3). *These are additional results extending Figs. 2 to 4 and 12 to 14*

# D   Experimental details

## D.1   Networks

**SimpleCNN:** SimpleCNN is a 6 layer fully convolutional neural network. Each convolution has a 3x3 kernel, stride 1 and bias. The layers have 32, 32, 64, 64, 128, and 128 channels from the first to the last layer. The weights are initialized using Kaiming initialization [37] and the biases are initialized to 0. There is a 2D maxpooling with a 2x2 kernel and stride 2 after layers 2, 4, and 6. Layer 6 is followed by a 2d global average pool which results in a 1x128 unit feature vector from which the classes are linearly predicted.

**ResNet20:** We use the ResNet20 used with CIFAR-10/100 data in the original paper [38].

**ResNet20 without batchnorm (RN no BN):** Same as ResNet20 but with batchnorm turned off.

Figure 18: Taylorized training versus ordinary training for ResNet20 without batch norm and trained with Adam on CIFAR-10 and CIFAR-100. Compared to Fig. 8, we included the second order Taylor expansions of the neural network as well as the first order. While the second order leads to lower error whenever we spawn in along the training trajectory, it still preserves the characteristic shape of the curve for the first order, where the training of the Taylorized model only starts getting better at a particular epoch. The dashed red line shows the performance of the neural network at epoch 100 which is not its peak performance yet, since it would still improve past this point.

Figure 19: Linearized training versus ordinary training for WideResNet on CIFAR-10 and CIFAR-100. Similarly to Fig. 8, we observe that the error of the trained linearized model retains the same shape, where it only starts getting better after a particular epoch. In contrast, the accuracy of the original model with the trained linearized model's weights plugged back in does not improve at all, suggesting that the WideResNet low loss basins might not be geometrically captured as easily by the linearized approximation. The dashed red line shows the performance of the neural network at epoch 100 which is not its peak performance yet, since it would still improve past this point.

| Network | Dataset | Opt | LR | Mom | WD | LR Decay | Decay Epochs | Total Epochs |
|---|---|---|---|---|---|---|---|---|
| Resnet20 | CIFAR10 | SGD | 1e-1 | 0.9 | 1e-4 | 0.1 | 80, 120 | 160 |
| Resnet20 | CIFAR100 | SGD | 1e-1 | 0.9 | 1e-4 | 0.1 | 60, 120, 160 | 200 |
| SimpleCNN | CIFAR10 | Adam | 1e-3 | - | 1e-4 | - | - | 100 |
| RN20 no BN | CIFAR10 | Adam | 1e-3 | - | 0 | 0.1 | 80, 120 | 160 |
| RN20 no BN | CIFAR100 | Adam | 1e-3 | - | 0 | 0.1 | 80, 120 | 160 |
| WRN-16-4 | CIFAR10 | SGD | 1e-1 | 0.9 | 5e-4 | 0.2 | 60, 120 | 160 |
| WRN-16-4 | CIFAR100 | SGD | 1e-1 | 0.9 | 5e-4 | 0.2 | 60, 120 | 160 |

Table 1: Training details for different experiments. For the Adam optimizers, the default hyperparameters were used. There was no learning rate decay for SimpleCNN.

**WideResNet-16-4 (WRN-16-4):** We use the WideResNet-16-4 described in [39] (16 layers, widen factor 4, no dropout).

## D.2 Training Details

See Table 1 for training details and hyperparameters.

## D.3 Extended training time and learning rate ablations

In Figs. 8 and 9, we train networks with a lower learning rate to compare to linearized training and measure the nonlinear advantage. This is done as follows: we first train a parent network at a constant learning rate of 0.1. At certain epochs (the x-axis of Fig. 8 and labels of Fig. 9) we spawn a child network and train it with a learning rate of 0.001 and independent SGD noise until convergence (the learning rate is chosen as the smallest learning rate at which the network converges in a reasonable amount of time, 1000 epochs in our case.) We then use

the learning rate dropped accuracy is the final accuracy that the child converges to. For stability, we repeat this process and average the two independent runs. The linearized training is performed using the Novak et al. [40] package and implemented in JAX [41]. It trains for additional 200 epochs at learning rate 0.001 that we choose based on a small-scale grid search for linearized training specifically. The nonlinear advantage is computed as the error difference between the final test performance of the low learning rate child network, and the final test performance of the linearized neural network.

## D.4   Linearized training details

The linearized training was performed using the Taylor expansions tools described in Novak et al. [13] and implemented in Novak et al. [40]. We trained for 200 epochs using SGD with Momentum, implementing the training loop in JAX [41]. The learning used was 0.001. We did a small-scale grid search for the best performing learning rate between 0.0001 and 0.1, choosing 0.001 in the end.