[Reviews · NeurIPS 2020]

Review 1

Summary and Contributions: The paper provides experimental evidence on CIFAR-10 and CIFAR-100 that the deep network determines the final basin of low loss within an epoch, and NTK corresponding to a trained network (only 10% of total training) can outperform full network training.

Strengths: - The paper defined numerous measurement metrics as tools to study the landscape. - The definitions in this paper are clear. The visualizations are good. - The topic is clearly relevant to the NeurIPS community.

Weaknesses: - The figures end with epoch 10^2, which is not the end of the training as mentioned in the appendix. Clearly the training error is not 0 at epoch 100 as shown in the figure, which means the training is not finished. This makes me doubt the conclusion that NTK corresponding to trained network can outperform full network training. - The paper only runs tests on CIFAR-10 and CIFAR-100. These two datasets are all image datasets and pretty similar. The results on other datasets (e.g., language dataset) can be very different. - The paper cannot draw any solid quantitative conclusions. The qualitative phenomenon has been known.

Correctness: It seems the neural network is not trained to have zero training error in the paper.

Clarity: The paper is overall well written.

Relation to Prior Work: The paper gives enough related works.

Reproducibility: Yes

Additional Feedback: Why the figures only show the first 100 epoch? ---------- post author response comments -------- The authors have addressed my main concerns of this paper (misleading information in Figure 6 and related section). I highly suggest that the authors do a major modification, especially on the conclusion. In addition, I agree with other reviewers' opinions that this paper conveys an important message to people that are not aware of this phenomenon. Therefore, I increased my score and happy to see this paper appearing in NeurIPS if the authors modify the paper accordingly (which I assume they will).


Review 2

Summary and Contributions: This work gives a large-scale phenomenological analysis of training based on various metrics related to the loss landscape and data-dependent NTK. The experimental results clarify that training dynamics within an epoch determine the final basin. After the basin fate is determined and within 10 epochs, data-dependent NTK acquires sufficient information to decrease the training and test error as a Talyorized model.

Strengths: *This work gives a unified and more sophisticated visualization of various well-established metrics that have been considered independently. In particular, the sophisticated evaluation of the parent-child spawning reveals that the decision of basin fate happens within 1 epoch. The parent-child spawning also leverages the evaluation of other metrics. *As a novel metric, this work investigated the approximation accuracy of Taylorized models with data-dependent NTK. It reveals that this NTK acquires sufficient information within 10 epochs. *These empirical findings will give rich insight and suggestions for further developing the theory of loss landscape and NTK.

Weaknesses: *The method of parent-child spawning [6] is not original to this paper. Besides, as remarked by authors, the early decision of final minima has also been argued by some other studies [9,10]. One can even easily imagine this phenomenon from the studies of mode connectivity [3,4]. Therefore, this paper's pure contribution seems limited to the evaluation of Talyerized models (that is, Fig. 6). *Some ambiguous explanation of experimental settings may prevent readers from correctly understanding the main claim. I will discuss it in the "Clarity" part.

Correctness: I find no incorrent description or methodology in this work.

Clarity: In Fig. 6, the linearized model with the data-dependent NTK out-performed the original network in the final phase of training. This is interesting but seems non-trivial (or strange) in the following two points. (i) The rightmost point in each figure appears to be given at 90 or 95th epoch. The rightmost green point takes lower errors than the rightmost red point. What happens if you take both points very close to 100th epoch (e.g. 99.99 epoch, 100 epoch minus 1 "step" )? I guess that the green point should be close to red one because the parameter update (1) should be not so large in the final phase (due to the small gradient, or scale learning rate by hands or Adam) and the Taylor approximation (2) will become more accurate. It sounds highly non-trivial that the green point always out-performs red one even at epochs very close to 100. (ii) The same problem also appears in the blue points. If you take the blue point sufficiently closer to 100th epoch, the blue point will become closer to the red one because both will share almost the same parameter. It would be necessary to confirm the consistency of the obtained results in a more thoughtful way arournd 100th epoch. One possible approach is to depict figures corresponding to Fig. 6 with the x-axis of [100th epoch, 100th epoch - 1 step, 100th epoch - 2 step, ...]. I believe this unclear point is not a serious flaw. But, if this point keeps ambiguous after rebuttal, I might be forced to decrease my score.

Relation to Prior Work: As I remarked in the above, the parent-child spawning is an established idea [6]. It would be better to discuss the difference between the current work and [6] more carefully. Certainly, the previous work [6] would not have been dealing with the function space. However, one can see the basin fate just by seeing the parameter space (Fig. 5 (left two)). Fig. 4 (right far) suggests that the tangent place in the function space is insufficient to represent the error. In Figs 4 and 5, is there any positive advantage to use the evaluation in the functional space compared to the previous studies in the parameter space [3,4,6]?

Reproducibility: Yes

Additional Feedback: Minor issues *Visualization method of Fig. 5: I am not sure how the authors depict this paper. Is it based on PCA of trajectories? It is also unclear why linear lines give these trajectories. *Line 21 "a kernel machine can actually out-perform...": The term "kernel machine" is misleading because authors do not perform kernel regression. It is just a linear regression with the Taylorized model (2). More technically speaking, when we use data-dependent NTK in a linearized model, the positive definiteness of this NTK is non-trivial and the equivalence to the kernel regression becomes unclear. *The term "chaotic": Authors sometimes refer to the training process in the early times as ''chaotic'', but does this mean literally "chaos" as a technical term? Is there any evidence for an exponential sensitivity against the initial condition? *The batch size used in experiments seems not mentioned in the manuscript. Batch size is known as a key factor in determining the shape of global minima [24]. It would be better to add any discussion on the dependence of the obtained results on the batch size. * Line 213 and Section B: The escape threshold has already been discussed in the context of neural networks [LeCun et al. "Efficient backdrop", 1998]. In the context of modern deep learning, [Karakida et al. https://arxiv.org/abs/1806.01316] revisited the same quantity and obtained its generalized form for the training with momentum. It would be better to refer to them. *Authors showed some experiments without batch norm in Supplementary Material, but not mentioned in the main text. Is there any discussion on the effect of batch norm? Batch norm is reported as a key factor in making the loss landscape smoother ([Santurkar et al. https://arxiv.org/abs/1805.11604]). Line 381: Reference [33] is missing in the main text. ===After rebuttal=== Thank you for your kind reply. Only after reading Reviewer 1's comment and your reply, I realized that total epochs were not 100. Since the current Fig. 6 (i.e., comparison with NN of epoch 100) causes misunderstanding, I recommend the authors to replace this figure with a new one shown in the rebuttal (i.e., comparison up to epoch 200). This new figure is insightful enough as a measurement of the geometric structure of training. I agree with other reviewers that the authors should do a major modification on some explanations regarding Figure 6. For instance, "within 10 percent of training time, the learned kernel as a kernel machine can actually outperform full deep learning" (which appears in abstract and conclusion) is highly misleading. It does not outperform the NN model of epoch 200. Although there are not a few inadequate descriptions, they do not flaw the significance of this paper, and I keep my overall score. I am looking forward to seeing a fully revised version.


Review 3

Summary and Contributions: This paper empirically studies many quantities related to the neural tangent kernel, including the dynamics of the kernel, the loss landscape, etc. It is claimed that after roughly 1 epoch of chaotic change in the learned kernel, the final basin containing the end point of training is basically determined. Then using the kernel learned after roughly 10 percent of the total training time, we can achieve good training and test errors.

Strengths: I think this paper provides some insights. For example, the kernel becomes stable after a long enough training (Figures 2 and 3, C-E), and the learned kernel after only 10 percent of the total training time can already achieve pretty good training/test error (Figure 6).

Weaknesses: I think more details and discussion are needed to justify and evaluate some claims in the paper. 1. In Figure 5, it is said that "two children spawned at an early time t_s in the chaotic training regime arrive at two different loss basins", while "two children spawned at a later time t_s in the stable training regime arrive at the same loss basin". While I do agree with this claim, a simple explanation for it may just be that as t_s becomes larger, the distance between a child and its spawning parent becomes smaller and smaller. To see this, note that as the risk decreases, the gradient norm also becomes smaller, and the child is trained for T-t_s epochs, which also shrinks as t_s becomes larger since T seems to be fixed. Therefore we can expect the child and parent to eventually lie in the same basin as t_s becomes larger, and I wonder what is the new conclusion we can draw from this experiment. 2. In Figure 6, at the last points of the curves, is \tilde{t} equal to T? If so the green curves should meet the red curves since there is no additional training, while if not I wonder the details. Regarding the claim that the learned NTK can outperform full deep learning, it seems that the kernel can indeed drives the training error smaller, but the improvement in test error is very small and may be negative as in Figure 12. I also wonder if the curves in Figures 6, 11 and 12 represent the average of multiple parent and child runs?

Correctness: Discussed above.

Clarity: I think the paper is well-written, but more details would be helpful, as discussed above.

Relation to Prior Work: The discussion of prior work is thorough as far as I know.

Reproducibility: Yes

Additional Feedback: Reply to the feedback: Thanks for the response! I agree it is interesting that the weights further travel a distance of 43 after epoch 30 while the learned NTK at epoch 30 is already able to give a good test accuracy. I think such details should be mentioned in the paper. On the other hand, I still think the current empirical results are not enough to conclude that the learned NTK can outperform full deep learning, particularly when considering the test accuracies. More discussion on this would be interesting.


Review 4

Summary and Contributions: The paper summarizes and compares three areas that aim fora better theoretical understanding of deep neural networks. In detail local, global loss landscapes as well as neural tangent kernels (NTK). Based on that the authors establish several criteria to measure how networks compare in terms of loss landscape and NTK. They are used to see if a child networks take at some point during training ("spawning" a network) diverge after further training. The results suggest that the first (5) epoch(s) are determining the final "loss basin" and how the network will converge. Furthermore, the authors show that a data-dependent NTK can trained to good accuracy (compared to a randomly initialized one). ---- read the reviews and the rebuttal and want to keep my score ----

Strengths: * Interesting results that likely contribute to the discussion in this field. * The empirical evaluation seems to be extensive.

Weaknesses: (I'm afraid I feel not familiar enough with the literature to give good critique here.)

Correctness: I am not familiar with the literature, but the paper makes a good impression. Experiments are outlined and detailed well and I would assume one could

Clarity: Yes, fairly easy to read.

Relation to Prior Work: I am not familiar with the literature, but the handling gives a good impression.

Reproducibility: Yes

Additional Feedback: As mentioned I am not too familiar with the literature in this area, but to not waste the review: My educated (phd in ML) opinion is that the submission is solid and by reading it twice it seems sound to me.

[Author Response · NeurIPS 2020]



Taylorized training with errorbars. Insets zoom in on early period of training.

Dear reviewers, thank you for your comments! We are happy to see that our empirical findings provide rich insight and
suggestions for further developing the theory of loss landscape and NTK (R2), that our metrics are numerous (R1), our
empirical analysis extensive (R6), and our results interesting and likely to contribute to the discussion in this field (R6).

**R3 :** To produce uncertainty estimates, we have now re-run Taylorized training experiments multiple times and **added**
**standard deviation error bars, which turn out to be very small**. An example is shown in the Figure attached.

**R1 :** We **added training epochs until convergence at epoch 200**. NTK (Taylor 1) outperforms full network training
in terms of training error, when both the full network and the NTK are trained up to 200 epochs. However, we found
that this lower training error for the NTK comes at the expense of overfitting, resulting in higher test error for the NTK.
Therefore, in our new analysis up to 200 epochs, we terminated NTK training based on the optimal **early stopping time**
**as determined by the test error of NTK**. Under this reasonable condition, we found that both NTK training and test
performance (green lines) almost matches that of the full network's test error at 200 epochs (dashed purple line), using
data-dependent NTK kernels that were created from the full network at 30 epochs for CIFAR-10 and 100 epochs for
CIFAR-100 (see Figure - compare green curves to dashed purple line). Notably, the NTK that is used at initialization,
and all fractional epochs less than 1, cannot be used to obtain test errors anywhere close to that of the full network at
200 epochs. However, within 3 epochs, we obtain an NTK that does significantly better than the one at initialization.

**R1 :** In terms of conclusions, these results provide important information to the field, namely that: (1) **NTK kernels**
**at initialization perform quite poorly**, but (2) **such kernels created after very few epochs of training perform**
**substantially better than at initialization**, and finally, **within 30-90 epochs of training these kernels have enough**
**information about the data in them to closely match the test error of the full network** at 200 epochs of full training.
We believe this first analysis of the dynamics of a data-dependent NTK kernel and its relation to landscape geometry will
be very useful for understanding relations between deep learning, NTK kernels, and geometry. Especially important is
the rapid improvement of a data-dependent kernel relative to initialization with less than 2 epochs of training (compare
green points at 2 epochs to green points at 0 epochs in the insets).

**R3 :** To falsify the hypothesis that children spawned late are in the same linearly connected mode because they don't
travel as far after being spawned, we measured the distance traveled in both cases. In Figure 5 of our paper, the L2
distance travelled from parent to trained child is **58 for spawn at init** and **43 for spawn at e=30**. While $58 > 43$,
this difference is not large enough to account for the effect. Therefore a **small distance travelled is likely not the**
**explanation** for the effect we see. Moreover, if we train children for a full $T$ epochs after spawn time $t_s$, rather than
$T - t_s$, we see similar effects. Therefore we conclude that it is the effect of the spawning time, not the length of the
subsequent optimization, that keeps late spawned children in the same basin.

**R6 :** *It sounds highly non-trivial that the green point always out-performs red one even at epochs very close to 100*:
The final accuracy of the Taylorized regime training depends on a) at what epoch you spawn it = develop the Taylorized
approximation, b) how well the Taylorized training works, and c) the existence of good optima within the Taylorized
approximation. While it is not a priori obvious, we found the green curve (error of trained Taylorized model) to be
lower than the red curve (original NN) at 100. However, at (newly added) epoch 200, this was still the case for train
but not test – training error went down, but overfitted and caused test error to go above the red line. This might be
due to the difficulty of training the Taylorized regime and our sub-optimal choice of hyperparameters. The Taylorized
model spawned at epoch $e$ gets additional training compared to the NN there, which is why the green curve can even *in*
*principle* be lower than the red curve.

**R1 :** We focused on image datasets as they are still a very important and prominent part of ML research. While we show
results for CIFAR-10 and CIFAR-100, we also ran experiments on MNIST, Fashion MNIST and SVHN, with equivalent
results. Due to the computational requirements of our experimental sweeps and Taylorized training in general, we
couldn't extend our analysis to ImageNet scale. However, we did use powerful models such as WideResNet and SOTA
training schedules and HP choices to make sure we can get as close to real settings as possible. We train to comparable
train and test error (see He et al.'15 for ResNet20 and Zagoruyko et al. '16 for WRN).

[Meta-Review · NeurIPS 2020]

The reviews for this paper were overall positive. The paper presents a empirical inquiry of the landscape of the loss of the data-dependent neural tangent kernel. The authors examine dynamics of the kernel, the loss landscape, comparing the learned kernel to corresponding neural networks. The reviewers appreciated that the evaluation of the so-called 'parent-child spawning' phenomena and the approximation accuracy of data-dependent neural tangent kernels. The authors made a laudable effort to put to question common preconceptions about neural tangent kernels through an extensive set of numerical experiments leading to interesting empirical observations. We recommend to carefully read the reviewers' comments and suggestions and take them into account while preparing the camera ready final version. The authors may pay particular attention to correct several misleading terms and to nuance several claims. If the final version is sufficient nuanced and polished, the paper will be a important and timely contribution to the field. Accept.